# Novel CaLB-like Lipase Found Using ProspectBIO, a Software for Genome-Based Bioprospection

**DOI:** 10.3390/biotech12010006

**Published:** 2023-01-06

**Authors:** Gabriela C. Brêda, Priscila E. Faria, Yuri S. Rodrigues, Priscila B. Pinheiro, Maria Clara R. Nucci, Pau Ferrer, Denise M. G. Freire, Rodrigo V. Almeida, Rafael D. Mesquita

**Affiliations:** 1Departamento de Bioquímica, Instituto de Química, Universidade Federal do Rio de Janeiro, Avenida Athos da Silveira Ramos 149, Rio de Janeiro 21941-909, Brazil; 2Department of Chemical, Biological and Environmental Engineering, Universitat Autònoma de Barcelona, 08193 Bellaterra, Catalonia, Spain

**Keywords:** bioprospection, CaLB, protein, functional characterization

## Abstract

Enzymes have been highly demanded in diverse applications such as in the food, pharmaceutical, and industrial fuel sectors. Thus, in silico bioprospecting emerges as an efficient strategy for discovering new enzyme candidates. A new program called ProspectBIO was developed for this purpose as it can find non-annotated sequences by searching for homologs of a model enzyme directly in genomes. Here we describe the ProspectBIO software methodology and the experimental validation by prospecting for novel lipases by sequence homology to *Candida antarctica* lipase B (CaLB) and conserved motifs. As expected, we observed that the new bioprospecting software could find more sequences (1672) than a conventional similarity-based search in a protein database (733). Additionally, the absence of patent protection was introduced as a criterion resulting in the final selection of a putative lipase-encoding gene from *Ustilago hordei* (UhL). Expression of UhL in *Pichia pastoris* resulted in the production of an enzyme with activity towards a tributyrin substrate. The recombinant enzyme activity levels were 4-fold improved when lowering the temperature and increasing methanol concentrations during the induction phase in shake-flask cultures. Protein sequence alignment and structural modeling showed that the recombinant enzyme has high similarity and capability of adjustment to the structure of CaLB. However, amino acid substitutions identified in the active pocket entrance may be responsible for the differences in the substrate specificities of the two enzymes. Thus, the ProspectBIO software allowed the finding of a new promising lipase for biotechnological application without the need for laborious and expensive conventional bioprospecting experimental steps.

## 1. Introduction

Enzymes are specialized proteins that catalyze chemical reactions and are therefore called biocatalysts. In general, biological reactions occur under mild conditions of temperature and pressure. In addition, the high specificity enables the reduction of co-products and waste. These advantages, and other factors that depend on the selected process, make biocatalysis in accordance with many of the 12 core principles of “green chemistry”, making several processes less environmentally damaging [1].

The discovery of new enzymes has thus been of great interest. Conventional bioprospecting methods involve the screening of enzyme-producing organisms, mainly microorganisms, by exploring natural sources such as industrial waste or soil. This process involves growth-dependent procedures and biochemical screening, which is usually time and resource consuming with low success rates. After these screening assays, the selected producer organism further needs to be identified, followed by the identification of the enzyme-coding gene sequence [2].

In this context, the contribution of bioinformatics to the bioprospecting field has proven to be a key driver in enzyme discovery, after advances in DNA sequencing and metagenomics [3]. As an example, the National Center for Biotechnology Information (NCBI—GenBank) currently lists annotated genomes from more than 2700 different fungal species. The growing number of deposited genomes are a consequence of the dramatic decrease in the sequencing cost in the recent years [4]. The post-genome analysis includes gene prediction, annotation, and submission to large databases such as GenBank [5], allowing easy search for genes and coded proteins using keywords or based on sequence similarity, such as BLAST (Basic Local Alignment Search Tool) [6]. Unfortunately, bioinformatic data processing is still challenging as existing software does not cover the many steps involved in the computational bioprospection, namely gene prediction, conserved domains, and functional annotations. A review focused on metagenomic bioinformatics tools for bioprospection were not able to cite a single software covering all these steps [7]. The gene prediction step is especially challenging [8], leading to many genomes being deposited in the databases without processing (gene prediction and annotation) which limits its usability.

As a case study, the present work invested in a search for a novel lipase considering the studies and commercial uses of these biocatalysts in recent decades for several industrial applications in the pharmaceutical, chemical, detergent, food, textile, baking, cosmetic, paper, biodiesel, and waste-treatment industries [9]. Recently, a technological forecasting study revealed that the use of lipases as biocatalysts is still important, and further research is needed for these enzymes to reach their full potential in the third and present wave of biocatalysis [10].

One of the most studied lipases for biotechnological uses is lipase B from *Moesziomyces antarcticus* (CaLB, after the previously named *Candida antarctica*) [11,12], which is currently marketed by Novozymes. CaLB is also active in non-aqueous organic solvents [13], making it possible to perform reactions such as esterification, transesterification, alcoholysis, and acidolysis [14,15,16,17]. Another potential feature is the high enantioselectivity towards secondary alcohols that is required for many pharmaceutical and chemical applications [18].

Robert and coworkers [19,20,21] accomplished the constitutive expression of the CaLB gene in *P. pastoris* and produced the so-called rLipB enzyme by using the 3-phosphoglycerate kinase (PGK) promoter. The rLipB production was carried out with crude glycerol as the substrate, with promising results, reaching 45 g/L of biomass and Y_P/X_ of 561 U/g without complete optimization [19]. This crude glycerol is a by-product of biodiesel production, reinforcing the economic potential of *P. pastoris* as a suitable host for producing microbial enzymes [22], including CaLB. Many studies have investigated novel lipases with a wide range of potential applications, such as CaLB. Uml2 and PlicB are two novel CaLB-like lipases prospected from *Ustilago maydis* and *Plicaturopsis crispa*, respectively, by conserved motifs and sequence homology to CaLB [23,24].

In the present study, we report the in silico prospecting of a novel CaLB-like lipase from fungal genomes. To accomplish this, ProspectBIO software was selected as a proof of concept as it presents advantages such as sequentially completing gene prediction and conserved domain annotation/filtering steps. The lipase-encoding gene from *Ustilago hordei*, a barley smut fungus, was chosen from many candidates after a patent search based on BLAST, then cloned and expressed in the methylotrophic yeast *Pichia pastoris*. A preliminary biochemical characterization of the corresponding enzyme, termed UhL, in comparison with rLipB was also performed. Even though the two enzymes display a high degree of homology, functional analyses indicated some differences between UhL and rLipB.

## 2. Materials and Methods

### 2.1. Bioinformatics

Lipase sequences from the superfamily abH37 (CaLB-like) from the Lipase Engineering Database (LED) [25] were obtained and aligned with MAFFT version 7.310 [26]. The alignment was used to construct a Hidden Markov Model (HMM) using hmmbuilt from the HMMER package [27] using the standard parameters.

The ProspectBIO software was developed (IP registry BR512018000940-9) and used for bioprospection, with the following parameters: (1) the taxon “Fungi” was chosen to allow the software to automatically download all fungal genomes available at the NCBI GenBank and RefSeq databases (ftp://ftp.ncbi.nlm.nih.gov/genomes/genbank/ and ftp://ftp.ncbi.nlm.nih.gov/genomes/refseq through 1 August 2019); (2) the CaLB protein FASTA sequence (acc. P41365 in the UniProt database) was set as the input sequence for gene prediction; and (3) the constructed CaLB HMM was used to filter and only keep sequences with the full conserved domain. These sequences were also annotated as CalB-like sequences.

For a comparative search, we performed an online protein BLAST (Basic Local Alignment Search Tool) hosted at NBCI (https://blast.ncbi.nlm.nih.gov/Blast.cgi#, accessed on 15 August 2019) setting CaLB as the query, using the “Non-redundant” (NR) database, and selecting only the Fungi taxon (taxid:4751). All standard parameters were maintained, except for the maximum target sequences which was increased to 5.000, the highest number available.

All identified sequences were compared with the CaLB sequence using local BLAST [6] and those with more than 70% sequence identity were followed-up with a patent search. These sequences were compared using BLAST in a local server against the “Non-redundant Patent Sequences—Protein” databases [28] from EMBL-EBI (European Molecular Biology Laboratory, European Bioinformatics Institute). The identified patents were manually evaluated, selecting those related with lipase use claims. The sequences related to these patents were disregarded.

All identified sequences were aligned with MAFFT version 7.310 [26] together with the CaLB sequence, using the default parameters. The alignment was converted to PHYLIP (.phy) format in the BioEdit software version 7.2.5 [29] and was manually verified. The phylogenetic tree was constructed based on a maximum likelihood tree using the JTT matrix model with RAxML software version 8.2.4 [30] with default parameters and 500 bootstraps. The tree was visualized with the iTOL (Interactive Tree Of Life) online tool 6.1.1 [31].

The isoelectric point (pI) and molecular mass were calculated using the ExPASy server. The protein three-dimensional model was created using the SWISS-MODEL server in the automatic mode, using the CaLB structure (PDB: 1TCA) as a template. The model quality was assessed by Errat [32] and Procheck [33] at the SAVES v6 server (https://saves.mbi.ucla.edu/, accessed on 13 January 2020). Images were generated by open-source PyMol v1.8.4.0 software.

### 2.2. Experimental

*Escherichia coli* DH5α was used for DNA manipulation. *Pichia pastoris* X-33 (Invitrogen-ThermoFisher, Waltham, MA, USA) was selected as the host for the gene expression. The pPICZαA vector (Invitrogen-ThermoFisher) was used for the synthesis of pPICZα-UhL by GenOne (Brazil). Zeocin and *Sac*I were purchased from ThermoFisher. DNA polymerase was obtained from Promega. Tributyrin and 4-methylumbelliferyl esters were purchased from Sigma-Aldrich. All other reagents used for this study were analytical grade. Lyophilized supernatant containing rLipB produced by *P. pastoris* was a gift courtesy of LaBiM (Laboratório de Biotecnologia Microbiana, Universidade Federal do Rio de Janeiro).

The DNA sequence encoding the *Ustilago hordei* putative lipase (GenBank: MN267804) was optimized for *Pichia pastoris* codon usage, synthesized, and cloned into *Xho*I and *Xba*I-digested pPICZαA vector using GenOne. The resulting plasmid, termed pPICZα-UhL, was transformed into *E. coli* and selected on Lysogeny Broth (LB) plates [0.5% (*w*/*v*) yeast extract, 1% (*w*/*v*) peptone, 0.5% (*w*/*v*) NaCl, and 2% (*w*/*v*) agar] containing 50 µg/mL Zeocin. Positive *E. coli* transformants were grown overnight in 10 mL LB medium containing 50 µg/mL Zeocin at 37 °C and 180 rpm. The pPICZα-UhL plasmid was isolated with a mini-prep kit (Qiagen, Venlo, The Netherlands) and linearized by *Sac*I digestion. About 10 µg of linearized pPICZα-UhL was transformed into 80 µL of electro-competent *P. pastoris* X-33 cells by electroporation according to Cregg and Russel [34]. After the transformation, the cells were grown on YPDS plates [1% (*w*/*v*) yeast extract, 2% (*w*/*v*) peptone, 2% (*w*/*v*) glucose, 1 M sorbitol, and 2% (*w*/*v*) agar] containing 100 µg/mL Zeocin for 48 h at 30 °C. About 10 transformant colonies were isolated and gDNA prepared using the LiOAc-SDS method according to Lõoke et al. [35]. A PCR protocol was used to confirm the gene insertion into the genome, using the following primer sequences: 5′-ttctccactggtaaagagattgcctca-3′ (forward) and 5′-gggttgcaatcagatgatctgtagtcct-3′ (reverse). In order to select recombinants with lipase activity, six isolated clones were grown on BMMY agar plates containing tributyrin [1% tributyrin, 1% (*w*/*v*) yeast extract, 2% (*w*/*v*) peptone, 1.34% (*w*/*v*) YNB, 100 mM potassium phosphate buffer pH 6.0, 0.4 mg/L biotin, 0.5% (*v*/*v*) methanol, and 2% (*w*/*v*) agar] for 4 days at 30 °C, with the addition of 250 µL of methanol on the plate lid every 24 h. The formation of tributyrin hydrolysis halos was monitored, and the strain with the highest hydrolysis halo was selected for further assays.

In order to produce the recombinant lipase, the selected strain was activated by inoculating 50 µL of a glycerol cryo-stock solution into 5 mL of YPD medium [1% (*w*/*v*) yeast extract, 2% (*w*/*v*) peptone, and 2% (*w*/*v*) glucose] containing 100 µg/mL of Zeocin in 50 mL conical tubes and grown overnight at 30 °C and 250 rpm. This culture was used to inoculate 250 mL shaker flasks containing 25 mL of BMGY medium [1% (*w*/*v*) yeast extract, 2% (*w*/*v*) peptone, 1.34% (*w*/*v*) YNB, 100 mM potassium phosphate buffer pH 6.0, 0.4 mg/L biotin, and 1% (*v*/*v*) glycerol] for a final OD_600nm_ = 0.1. After culturing at 30 °C and 250 rpm in a New Brunswick^TM^ Innova^®^ 42R incubator (Eppendorf) for 24 h, the cells were centrifuged at 2000× *g* for 5 min at 4 °C. The pellet was resuspended in 25 mL of BMMY medium [1% (*w*/*v*) yeast extract, 2% (*w*/*v*) peptone, 1.34% (*w*/*v*) YNB, 100 mM potassium phosphate buffer pH 6.0, 0.4 mg/L biotin, and 1 or 2% (*v*/*v*) methanol] in 250 mL shaker flasks and cultured at 20 or 25 °C and 250 rpm for 120 h. Methanol was added every 24 h at a final concentration of 1% or 2% as an inductor to maintain the recombinant protein expression. After culturing, the cells were centrifuged at 10,000× *g* for 15 min at 4 °C and the supernatants were stored at −20 °C until further analysis.

Deglycosylation of the supernatant containing the recombinant enzyme was carried out using endo-β-N-acetylglucosaminidase H (Endo H_f_, New England Biolabs, MA, USA) following the manufacturer’s instructions. In order to specifically observe the recombinant enzyme, a Western blot was carried out. sodium dodecyl sulfate poly-acrylamide gel electrophoresis (SDS-PAGE) was conducted, and the proteins were transferred to a nitrocellulose membrane using a Trans-Blot^®^ SD Semi-Dry Transfer Cell (Bio-Rad). After blocking with 5% skim milk overnight, the membrane was incubated with rabbit anti-HisTag antibody (Santa Cruz) for 2 h, washed several times with TBST buffer [20 mM Tris, 150 mM NaCl, and 0.05% Tween-20, pH 7.6] for 5 min and then incubated with alkaline phosphatase conjugated anti-rabbit antibody (Sigma-Aldrich, St. Louis, MI, USA) for 1 h. The bands were revealed by incubating the membrane with alkaline phosphatase substrates.

Activity was measured for supernatant samples containing UhL and rLipB by the spectrofluorimetric method using 4-methylumbelliferyl (MUF) esters as substrates. The activity assay was conducted according to Prim et al. [36], using 50–100 µL of the protein sample at a standard temperature of 45 °C in 0.05 M phosphate buffer pH 7 with MUF-heptanoate (MUF-7) as the standard ester substrate. One unit of enzymatic activity was defined as the enzyme quantity necessary to catalyze the formation of 1 µmol of MUF per min. Substrate preferences were evaluated using the following 4-methylumbelliferyl (MUF) esters: MUF-acetate (C2), MUF-butyrate (C4), MUF-heptanoate (C7), and MUF-oleate (C18). Activity assays were performed using these substrates, following the method described for MUF-7.

## 3. Results

### 3.1. Bioprospecting

#### 3.1.1. ProspectBIO Software

The ProspectBIO software (IP registry BR512018000940-9) provides searches in genomes for genes coding proteins with sequence similarity to an enzyme/protein of interest. It modularly manages the genome downloads, the similarity-based gene prediction and translation, and the protein sequence filtering by conserved domain (Figure 1). Initially, a group of genomes of interest is selected based on species or genus name or even taxonomic terms like “fungi”, “bacteria”, etc. They are downloaded from the NCBI GenBank and RefSeq databases without gene predictions. Subsequently, the input sequence is used as a reference by the EXONERATE program [37] to identify genes that encode homologous proteins in each downloaded genome followed by the creation of the coding (CDS) and protein FASTA sequences. To check the integrity of the predicted proteins and avoid the selection of low-quality or fragments, ProspectBIO optionally performs a search for a preset conserved domain (CD) using the HMMsearch program [27]. The user must have the CD HMM file in a Pfam-like format to allow the filtering of the predicted proteins having at least one copy of that CD. The user can identify the CD in the reference protein using InterPro (https://www.ebi.ac.uk/interpro/search/sequence/, accessed on 5 August 2019) and its Pfam HMM file must be manually downloaded at the “curation” tab of the CD entry (ex: https://www.ebi.ac.uk/interpro/entry/pfam/PF00001/curation/, accessed on 5 August 2019). Thus, the ProspectBIO final result is stored in a single FASTA file where each sequence is named with a code plus the EXONERATE score, the HMMsearch score, and the identity and similarity percentages compared to the reference sequence.

#### 3.1.2. Prospecting a Novel CaLB-like Lipase

The ProspectBIO software was used to prospect a novel fungal CaLB-like lipase. The dataset contained 4366 fungal genomes by August 2019; the gene prediction was based on the CaLB sequence, and the CD filtering used the data from the lipase superfamily abH37 of LED, finding 1672 CaLB similar sequences. On the other hand, the NCBI BLASTp search using the NR database and limited to the fungal taxonomic point only found 733 similar sequences.

A total of 30 ProspectBIO sequences presented with greater than 70% sequence identity with CaLB. These CaLB-like sequences were then blasted [6] with the “Non-redundant Patent Sequences—Protein” database [28] finding eleven patents. Seven of them (WO2011013707A1, WO2010124968A1, WO2009093703A1, WO2009077523A2, WO2009011354A1, WO02095127A2, and US5817490A) were discarded as the claims were outside of our research focus (they were about amino acid, statin or 2-keto-l-gulonic acid production, paper recycling, and enzyme separation methods). In contrast, four patents (WO2008065060A2, WO1993024619A1, WO2004024954A1, and JP2007300914A) were related to lipase activity, mostly to CaLB variants, but also CaLB-like lipases from *Hyphozyma* sp., *Ustilago maydis*, *Gibberella zeae*, *Debaryomyces hansenii*, *Aspergillus fumigatus*, *Aspergillus oryzae*, and *Neurospora crassa*. The most relevant claim among those patents protects sequences with lipolytic activity and more than 80% identity to the CaLB sequence (1TCA), citing a list of residues where the dissimilarity should occur. Therefore, the lipases from *Moesziomyces aphidis* (99%), *Moesziomyces sp* F5C1 (98%), and *Moesziomyces sp* F16C1 (98%) were disregarded.

The phylogenetic tree of the 30 prospected sequences with ID > 70% (Figure 2) showed two main groups (1 and 2), two ungrouped lipases (*Testicularia sp.* and *Pseudozyma sp*), and the three disregarded sequences due to the patent search. Representative sequences from the two groups are listed in Table 1. The most attractive CaLB-like sequence without patent protection seems to be the one predicted from *Ustilago hordei* (and also present in the GenBank with accession no. CCF54401) as it has one of the highest sequence identities to CaLB (76%). Notwithstanding, it comes from a well-known fungus with a greater number of studies reported in the literature when compared with the other species presented in Table 1. Its relevant to cite that the patent WO2017208791A1 cites an *Ustilago hordei* lipase sequence (Sequence ID 4, available in the South Korean version of this patent, KR20190015393A), however the sequence is very dissimilar (less than 10% identical) to the lipase studied here.

#### 3.1.3. UhL Three-Dimensional Modeling

Full-length UhL and the well-known lipase CaLB have 76% identity, so the three-dimensional model of UhL was calculated using the CaLB structure (PDB: 1TCA) as the template (Figure 3). Both enzymes also show the highly conserved pentapeptide Thr-X-Ser-X-Gly around the catalytic serine, as well as the six cysteine residues that occur in most lipases [38]. In addition, the conserved catalytic triad of the α/β hydrolase family is represented by the amino acids Ser111, Asp193, and His230 in the UhL sequence (Figure 3).

### 3.2. Experimental Validation

#### 3.2.1. UhL Expression in *Pichia pastoris*

In view of the discussion in the previous topics, the *Ustilago hordei* lipase (UhL) gene sequence prospected by the ProspectBIO methodology was chosen as a proof of concept. The experimental validation described below included cloning and expression in *Pichia pastoris*, as well as determining the substrate specificity of the enzyme obtained from the crude extract.

Tributyrin is a triglyceride widely used to detect lipase activity [39,40]. The reference strain X-33 and transformants harboring the UhL-encoding gene were grown in the presence of tributyrin substrate (Figure 4a) and the hydrolysis halos were shown for the positive recombinants. Activity was detected in the supernatant of the cultures after 144 h of fermentation (Figure 4b). Different induction conditions were tested, including the incubation temperature (20 and 25 °C) and inductor concentration (1 and 2% methanol). 

#### 3.2.2. UhL Characteristics

Expression of the cloned *U. hordei* lipase gene resulted in a 330 amino acid protein sequence with a carboxyl-terminus tag composed of six histidines. The protein molecular mass was predicted as 35 kDa and experimentally estimated to be 36 kDa by Western blot analysis (Figure 4c; lane 1) against the C-terminal histidine tag. This molecular mass was similar to the 34 kDa that was previously reported for CaLB [41]. However, the theoretical isoelectric point (pI) for UhL was predicted to be 9.3, which is very different from the theoretical and experimental values reported to be around 6.0 for CaLB [42].

The residue Asn80 was predicted to be *N*-glycosylated by the NetNGlyc 1.0 server, with a 0.7123 score, as it occurs in the tripeptide consensus sequence Asn-X-Thr/Ser. This site is probably glycosylated by *P. pastoris*, as glycosylation of UhL was confirmed by the decrease in molecular mass in the Western blot analysis after EndoH digestion (Figure 4c; lane 2). In CaLB, the structurally conserved Asn99 residue is also reported as glycosylated when expressed in *P. pastoris* and *Aspergillus oryzae* [41,43]. Unlike other lipases, the glycosylation appears to not influence the enzymatic activity of CaLB [43,44].

#### 3.2.3. Substrate Specificity

Different substrate specificities between the UhL and rLipB samples were observed by using 4-methylumbeliferyl (MUF) esters of different chain lengths as substrates (Figure 5). Both enzymes showed the highest activity with MUF-heptanoate (MUF-7); however, statistically significant differences were shown for the other acyl moieties studied. The rLipB activity was increased, as the ester substrate chain length increased up to C7 and decreased with ester chains longer than the intermediate range. This profile shown by the rLipB sample has also been reported for CaLB, using *p*-nitrophenyl esters as substrates [19,23,24,43]. In contrast, the UhL supernatant showed higher activity with C2 than with C4.

## 4. Discussion

Bioinformatics-based bioprospecting is a great strategy to search for new catalysts to develop or improve sustainable processes that includes gathering, curating, and analyzing database information. The BLAST tool is one of the most convenient software as the databases are online. The NCBI non-redundant (NR) and nucleotide databases, despite being the larger databases available in BLAST, do not contain all genome-coded sequences. On the other hand, the NCBI raw genome database that contains the most complete gene repertoire (despite uncovered) is not available online as a BLAST database. 

Genome-based bioprospection requires bioinformatic tools that are less convenient to use than BLAST, including the need to locally install many different software and databases, as exemplified by the review of metagenomic genome-based bioprospection [7]. Usually, the user needs one software for each of the many steps needed in a bioprospection analysis pipeline; as far as we know, no software is available to actually execute a pipeline containing many steps.

The ProspectBIO development aimed to fill the gap of more easy-to-use software that can execute many steps of the bioprospection altogether. Notwithstanding, it was also developed to make its own gene prediction, being able to use genome data directly, locally (using unpublished or private data), or downloaded from the NCBI GenBank and RefSeq databases. Its modular design allows the user to use any combination of (1) taxonomic-based genome download, (2) protein-based gene prediction on genomes and transcriptomes, and (3) HMM conserved domain filtering.

The ProspectBIO search for a CaLB-like lipase was tested on *fungi* genomes and found more sequences than a BLAST search with the same taxonomic limitation. There was an overlap between the results of both strategies; for example, the *U. hordei* lipase identified (UhL) has a protein accession number (CCF54401) that shows it is also present in the NR database.

The patent search based on sequence similarity used BLAST and was convenient but does not completely avoid the patent reading. The presence of a sequence in a patent does not mean that it is protected for a specific use and needs review by a human. The lipase exclusions after the patent analysis were restricted to sequences with very high identities to CaLB. One interesting point was that the excluded sequences did not contain accession numbers and were not present in the NCBI NR database.

The phylogenetic tree (Figure 2) showed branches enriched in a single or few genera, some of them have bootstrap values about 50%, which can be considered low but is compatible with the fact that all sequences in the tree are very similar. The *U. hordei* lipase identified belongs to Group 1 (*Ustilago* branch), which was supported by a bootstrap of 47%. The *U. hordei* protein sequence showed one of the highest sequence identities to CaLB (76%), but the 24% difference opens possibilities to obtain functional diversity between the two lipases. Therefore, scientific exploitation of a novel lipase from *U. hordei* could be of particular interest for the research already established for this fungus.

The identity between UhL and CaLB allowed the created three-dimensional model to be adjusted, with a considerable Qmean value of 0.16 [45] and an overall quality estimate (GMQE) of 0.92. The overall structure has a typical α/β hydrolase fold [46] (Figure 3a) with a central β-sheet composed of seven β-strands surrounded by α-helices. The catalytic residue of Ser111 is located at the well-known nucleophilic elbow [38] formed by a right turn between the β4-strand and the α4-helix, a common feature of most lipases including CaLB [13]. Disulfide bonds are present in the UhL structural model in similar locations as in CaLB (Figure 3a), totaling three disulfide bridges. The presence of the typical lipase characteristics strongly suggests that this sequence will have lipase activity when produced but the amino acids substitutions observed in the entrance of the active pocket (Figure 3) reinforce the possibility of differences regarding the substrate specificity.

UhL showed to be the most promising among the similar protein sequences to CaLB for the present study, and was further cloned and expressed in *Pichia pastoris*, showing activity against tributyrin triglyceride (Figure 4a). Small-scale expression studies in shaker flasks revealed that temperature and methanol concentration are important parameters influencing UhL production (Figure 4b). Temperature can affect the formation of disulfide bonds and exposure of hydrophobic surfaces. In *Escherichia coli*, higher temperatures favor the formation of disulfide bonds [47], which is often important to stabilize the protein structure. On the other hand, exposure of the hydrophobic surfaces occurs preferentially at higher temperatures, which favors protein aggregation [48]. In general, lowering the temperature reduces the stress response and the protein synthesis rate, allowing the nascent peptide chains to fold correctly [47,49]. In addition, expression at lower temperatures in *P. pastoris* has been reported as beneficial for the production of different recombinant enzymes, due to the decrease in extracellular proteolysis [50,51], as reported for the expression of polygalacturonate lyase at 22 °C [52]. 

The induction phase is also dependent on the methanol concentration; here, we demonstrated that for the UhL production, induction at a 2% methanol concentration increased the lipase activity levels in the culture supernatant from 21 to 43.7 U/L compared to 1% methanol. Similarly, for some other proteins such as irisin and a humanized antibody, 2 and 3% methanol concentrations were the best inducing conditions in *P. pastoris* shaker-flask cultures, respectively [53,54]. Conversely, for many other recombinant enzymes, lower concentrations show better results [55]. The use of a 0.5 to 1% methanol concentration is also recommended in the ThermoFisher manual [56], as higher concentrations can be toxic to the cells [57,58]. Nonetheless, induction conditions on a small scale are highly dependent on culture parameters, influencing methanol consumption and evaporation, such as the stirring rate and aeration, methanol pulse addition frequency, and shaker-flask or well-plate geometry, which may also explain the divergence of results.

Different substrate specificity profiles between the UhL and rLipB samples were observed (Figure 5) and this might be a result of the residue substitutions. The surface analysis of the UhL model in comparison with CaLB revealed differences in the entrance of the active pocket (data not shown). Two of the amino acid substitutions studied may have the capacity to also change the active pocket polarity, as Glu146 and Ser157 of UhL replaced Leu140 and Ala151 in the CaLB sequence, respectively. These replacements, as well as Leu155 and Leu291 from the UhL sequence, were the same as reported for Uml2, another CaLB-type lipase from *Ustilago maydis* [23]. Buerth and coworkers [23] observed a different profile of substrate chain-length specificity for CaLB against p-nitrophenyl esters in comparison with Uml2, which was similar to the UhL specificity profile against 4-methylumbeliferyl esters. Thus, the present work reinforces the importance of these amino acids substitutions for the substrate specificity observed for UhL and Uml2 when compared to CaLB. Further analysis could confirm these observations.

## 5. Conclusions

In conclusion, a new, easy-to-use bioprospection software named ProspectBIO was shown. It has many steps “in-a-box” including NCBI GenBank and RefSeq genome download, gene prediction based in a single input protein, and conserved domain filtering to retrieve potentially functional sequences. Its use may avoid expensive and laborious experimental approaches. The patent search performed after the prospection was an important strategy when it comes to products with industrial potential. The prospected *Ustilago hordei* lipase (UhL) has a sequence identity around 76% and was produced by heterologous expression in *Pichia pastoris*. Indeed, the functional characterization of the recombinant enzyme UhL showed differences in relation to CaLB in terms of substrate specificity. Amino acid substitutions presented in the active pocket may be responsible for some of these differences, bringing insights for future biotechnological applications of this novel lipase.

## 6. Patents

The ProspectBIO software was registered with the intellectual property code BR512018000940-9.

## Figures and Tables

**Figure 1 biotech-12-00006-f001:**
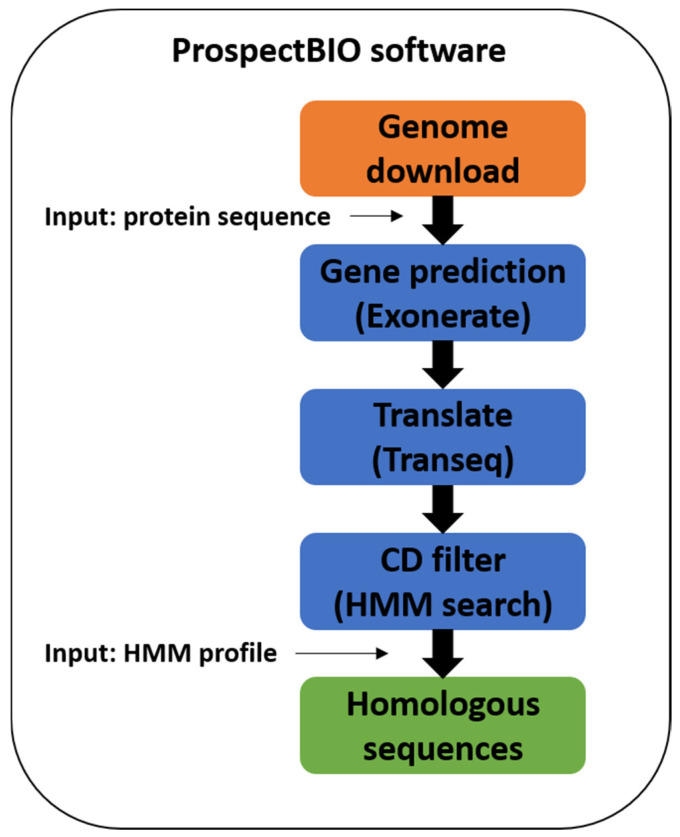
Prospecting methodology to obtain potential protein sequences using the ProspectBIO software.

**Figure 2 biotech-12-00006-f002:**
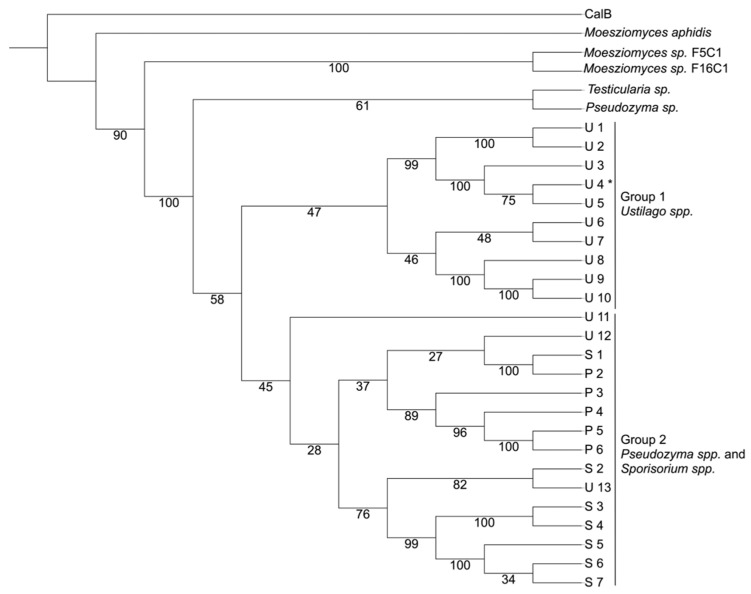
Phylogenetic tree of the prospected sequences with greater than 70% identity with CaLB (P41365). The groups were created selecting the branches where there were few species with high frequencies. Group 1—Ustilago spp.; and Group 2—*Pseudozyma* spp. and *Sporosorium* spp. The outgroup sequence was CaLB. The *Ustilago hordei* (CCF54401) is U4 and the *Sporisorium scitamineum* SSC39 (CP010917) is S6. The numbers represent the bootstrap values. * Lipase studied in this manuscript.

**Figure 3 biotech-12-00006-f003:**
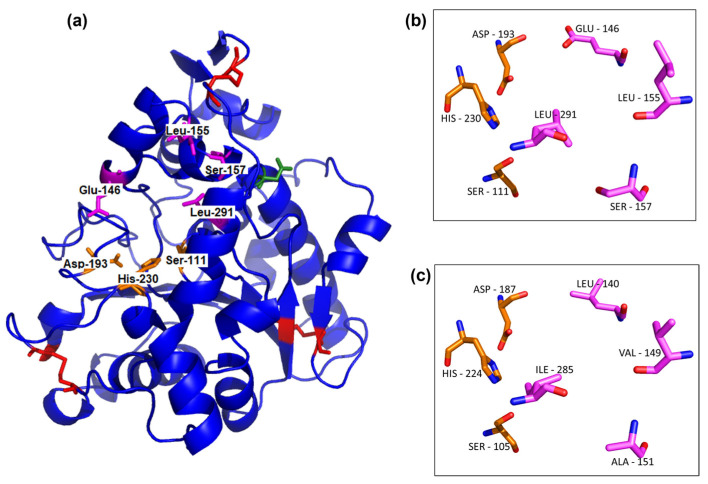
The UhL model was calculated by the SWISS-MODEL using CaLB (PDB number: 1TCA) as the template (Qmean = 0.16; GMQE = 0.92). Sticks in red are the disulfide bonds, in orange the catalytic triad (Ser111, Asp193, His230), in green the glycosylation site, and in pink the four amino acid differences between UhL and CaLB (**a**). The catalytic triad and four different amino acids in the active pocket are shown in detail for UhL model (**b**) and CaLB structure (**c**). The oxygen and nitrogen atoms are shown in red and blue, respectively, in panels (**b**,**c**).

**Figure 4 biotech-12-00006-f004:**
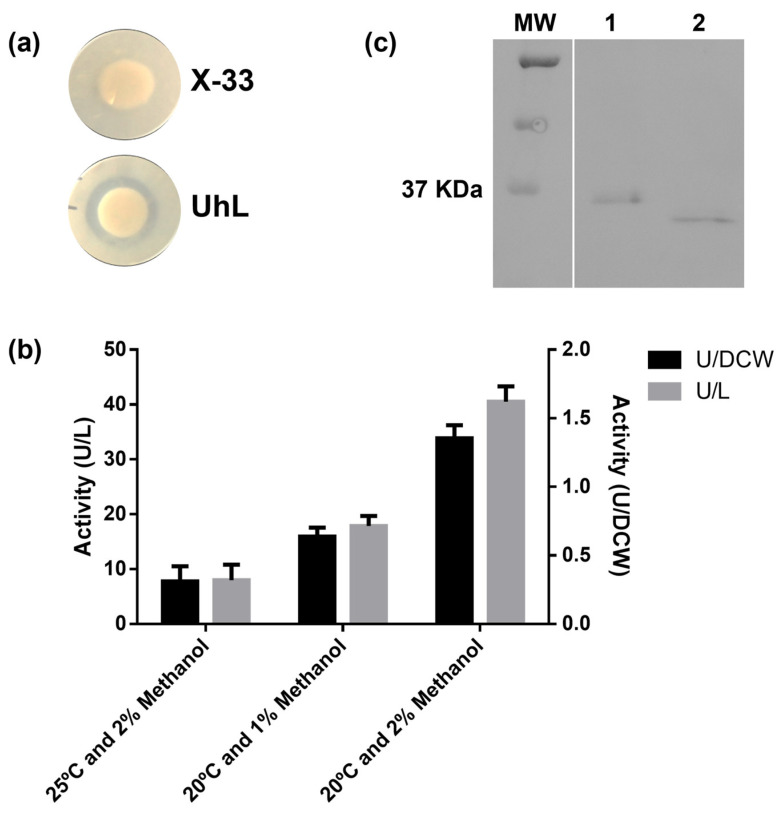
Activity towards tributyrin substrate on solid medium plates for the wild type (X-33) and mutant (UhL) strains after 4 days of incubation (**a**). Volumetric activity (U/L) and specific activity per dry cell weight (U/DCW) obtained in the supernatant after 144 h cultivation of the mutant under different induction conditions (**b**). Western blotting anti-HisTag of supernatant containing the recombinant lipase before (1) and after (2) treatment with EndoH (**c**). Molecular weight marker (MW). The western blotting full image is in Appendix A.

**Figure 5 biotech-12-00006-f005:**
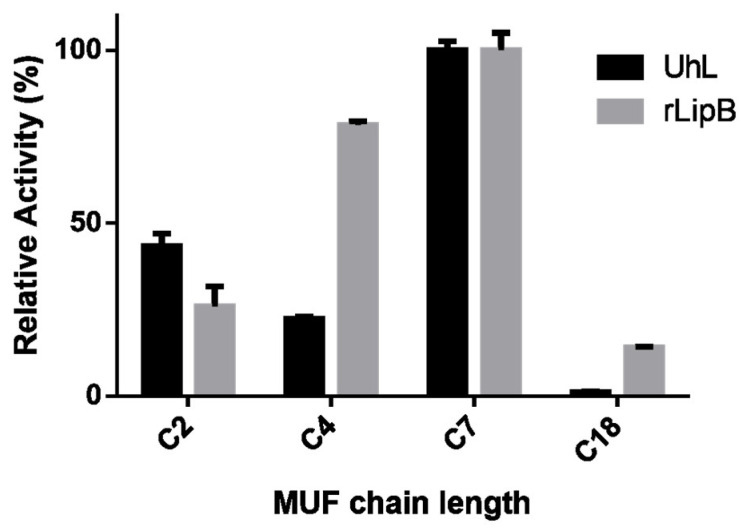
Relative activity of the enzymes UhL (black) and rLipB (gray) toward 4-methylumbelliferyl esters with different carbon chain lengths. The enzymatic assays were carried out at 45 °C and pH 7.0.

**Table 1 biotech-12-00006-t001:** Representative lipase sequences that were identified. The *Candida antarctica* lipase CaLB was included in the table for comparison. The identity percentage (ID%) is in referenced to the CaLB sequence.

Accession Numbers	Organism	Protein Length	ID%
ACI06118			
1TCA	*Candida antarctica*	342	100
P41365			
CCF54401	*Ustilago hordei*	343	76
CP010917	*Sporisorium scitamineum* SSC39	339	74

## Data Availability

The publicly archived datasets that were analyzed were the NCBI GenBank database (ftp://ftp.ncbi.nlm.nih.gov/genomes/genbank/, accessed on 1 August 2019) and the NCBI RefSeq database (ftp://ftp.ncbi.nlm.nih.gov/genomes/refseq, accessed on 1 August 2019). The NCBI “Non-redundant” (NR) database was used online by BLASTp (https://blast.ncbi.nlm.nih.gov/Blast.cgi# on 15 August 2019). The EMBL-EBI “Non-redundant Patent Sequences—Protein” databases were downloaded from https://www.ebi.ac.uk/patentdata/nr, accessed on 30 August 2019.

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
