# Peer review of "Novel CaLB-like Lipase Found Using ProspectBIO, a Software for Genome-Based Bioprospection"

_biotech, 2023, doi:10.3390/biotech12010006_

Round 1

Reviewer 1 Report

The article "Finding a novel CaLB-like lipase using ProspectBIO, a software 2 for genome-based bioprospection" by Gabriela Brêda and co-author can be accepted for publication in BioTech after the following modifications.

 (1) Please Change the title to MDPI BioTech format.

 (2) Please incorporate a concluding section for the novelty, accomplishment and future direction of this unique software based approach.

 (3) In line 54, please briefly explain the latest challenges involved in bioinformatic data processing and the limitations of existing software.

 (4) Please check all the abbreviations and mention the full name when it appears for the first time. E.g., in line 54, 'BLAST'. Make a throughout check for the common typos. E.g., in line 121, remove one 'and'.

 (5) Please correct the figure paréntesis for Figure 3 and Figure 4. Unify to caps or small, (A), (B), etc.

 (6) Please label the active pocket and amino acids in Figure 3A.

 (7) It could be beneficial if a more straightforward explanation could be given for the utility/connectivity of ProspectBIO software towards Prospecting CaLB-like lipase.     

Author Response

Dear reviewer,

Reviewer 2 Report

In this study, bioinformatics technology was used to find new biological functional enzymes, which were verified by experiments. The composition of the article is smooth, and the quality of the chart is also high.

However, there are also some problems.

1. Figures 3 and 4 are composed of multiple pictures, and a), b) and c) are more intuitive in the upper left corner of the picture.

2. The article has no conclusion?

3. What is the difference between Prospect BIO and the same type of software? What are the advantages of ProspectBIO BIO? It is suggested to add some expressions in the introduction and discussion.

Author Response

Dear reviewer, 
